# Influence of Increased Radiation Background on Antioxidative Responses of *Helianthus tuberosus* L.

**DOI:** 10.3390/antiox12040956

**Published:** 2023-04-18

**Authors:** Oksana B. Polivanova, Kirill N. Tiurin, Anastasia B. Sivolapova, Svetlana V. Goryunova, Sergey V. Zhevora

**Affiliations:** 1Laboratory of Cell and Genomic Technologies, Russian Potato Research Center, 140051 Kraskovo, Russia; 2Department of Biotechnology, Russian State Agrarian University, Moscow Timiryazev Agricultural Academy, Timiryazevskaya Str., 49, 127550 Moscow, Russia; 3Laboratory of Systemic Genomics and Plant Mobilomics, Moscow Institute of Physics and Technology, Institutsky Lane, 9, 141701 Dolgoprudny, Russia; 4Laboratory of Marker and Genomic Plant Breeding, All-Russia Research Institute of Agricultural Biotechnology, Timiryazevskaya Str., 42, 127550 Moscow, Russia

**Keywords:** antioxidant enzymes, low-molecular antioxidants, *Helianthus tuberosum* L., ionizing radiation

## Abstract

As a result of the accident at the Chornobyl Nuclear Power Plant, significant territories were exposed to ionizing radiation. Some isotopes, such as ^137^Cs, are capable of making a significant impact on living organisms in the long-term perspective. The generation of reactive oxygen species is one mechanism by which ionizing radiation affects living organisms, initiating mechanisms of antioxidant protection. In this article, the effect of increased ionizing radiation on the content of non–enzymatic antioxidants and the activity of antioxidant defense enzymes of *Helianthus tuberosum* L. was studied. This plant is widely distributed in Europe and characterized by high adaptability to abiotic factors. We found that the activity of antioxidant defense enzymes, such as catalase and peroxidase, weakly correlated with radiation exposure. The activity of ascorbate peroxidase, on the contrary, is strongly positively correlated with radiation exposure. The samples growing on the territory with constant low exposure to ionizing radiation were also characterized by an increased concentration of ascorbic acid and water-soluble phenolic compounds compared to the controls. This study may be useful for understanding the mechanisms underlying the adaptive reactions of plants under prolonged exposure to ionizing radiation.

## 1. Introduction

As a result of the accident at the Chornobyl Nuclear Power Plant (NPP), vast territories were contaminated with radionuclides. For the first few weeks, short-lived radioisotopes had the main radiation effect on populations, while in the long term, a significant contribution is made by the long-lived radionuclides of ^137^Cs, whose half-life is 30 years [1]. Due to the fact that higher plants live an attached lifestyle, they are forced to adapt to various environmental conditions, including salinization, drought, low and high temperatures, etc. Many of them cause stress and cause a specific response in plants—an adaptation syndrome that includes various protective mechanisms.

One of these stress factors for plants is ionizing radiation generated by radionuclides. Radiation emission leads to the formation of an excessive amount of reactive oxygen species (ROS) caused by an imbalance between their formation and destruction [2,3]. The accumulation of ROS leads to the development of oxidative stress, which is expressed in a change in the metabolism of plant cells, necrotic damage of organs, and even death. However, plants can maintain their viability through the presence of an antioxidative system (AOS) that neutralizes ROS. The AOS includes enzymatic and low molecular weight antioxidants, which are chemical compounds of different kinds, including secondary metabolites [4].

The evaluation of the absorbed dose required to activate adaptive mechanisms in plant populations in response to ionizing radiation remains the subject of discussion. The maximum dose rate which does not cause phenotypic changes and is considered to be safe for plants in natural populations is 0.4 mGy/h [5]. However, lower doses of ionizing radiation can create stressful conditions for plants. Studies of the effects of radiation exposure on plants are primarily conducted in the laboratory due to the simplicity of the experiment and the interpretation of the data obtained [3]. The research data demonstrates that low doses usually stimulate plant growth and development [6,7,8], while high doses can cause significant negative effects [9,10]. These experiments are reduced to exposing plants to an external radiation source and comparing the results with the control. At the same time, radiation exposure levels can significantly exceed any natural conditions [11].

Field experiments provide a more comprehensive view of the effects of radiation exposure. They involve studying the long-term effects of ionizing radiation on plant growth and development at the sites of nuclear accidents (nuclear accident sites), such as Chornobyl, and Fukushima, nuclear weapons testing sites, and areas with naturally high levels of radiation [12,13,14,15,16].

Essentially, this kind of study allows us to obtain unique data on the molecular mechanisms of adaptation to long-term anthropogenic impact and predict the state of plant populations growing in areas contaminated with radiation.

*Helianthus tuberosus* L. (Jerusalem artichoke), also known as sunchoke, is a perennial herbaceous *Asteraceae* family plant native to North America [17]. This species has become broadly distributed throughout the world [18]. Jerusalem artichoke is adaptable to different growing conditions and in various types of soils (sand, loamy sand, loam clay) [19] and can be cultivated in areas with distinct climate types [20]. The species is tolerant to abiotic stresses, such as drought, water stress, and salinity [19]. It can also be cultivated in a broad range of temperatures [21].

*H. tuberosus* has great potential for use in agriculture and industry. Its tubers contain abundant biologically active compounds and nutrients for the human diet, such as vitamins (group B and A), microelements (iron, calcium, potassium, phosphorus), and inulin [20].

*H. tuberosus* tubers is widely used as a raw material in the chemical industry for paper pulp production, acetone, methane, ethanol, butanol, lactic acid, propionic acid, mannitol, and pectic substances productions [22]. *H. tuberosus* is also grown as an ornamental plant and is widespread as a weed.

Due to the high adaptability and ubiquity of *H. tuberosus*, its studies could be interesting in terms of resistance to prolonged exposure to ionizing radiation.

The purpose of this research was to study the changes occurring at the biochemical level in *H. tuberosus* plants in response to growing in conditions of increased radiation background.

## 2. Materials and Methods

### 2.1. Characteristics of Experimental Areas and Plant Material

The study was conducted on a common variety of *H. tuberosus* called ‘Skorospelka’, which can grow throughout the entire territory of the middle latitudes. The experimental sites selected to study the adaptive reactions of *H. tuberosus* plants to increased radiation background are located in areas contaminated with radionuclides after the Chornobyl accident. The highest levels of pollution on the territory of the Russian Federation are observed in the Bryansk region, where two experimental sites were selected: the village of Rozhny (ROZH) and the village of Olkhovka (OLKH). Four uncontaminated sites (Moscow, Orel, and Kaluga regions) with radioactivity at the level of the natural background were selected as control sites: Maslovo village (Solnechnogorsk district; K1), Botovo village (Chernogolovka district; K2), Kolpna village (Oryol region; K3) and “Druzhba” SNT (Zhukov district; K4). The location of the experimental sites and their characteristics are shown in Figure 1 and Table 1.

### 2.2. Soil and Plant Sample Collection in Experimental Area

Samples of soils and tubers of *H. tuberosus* constantly growing on this territory were taken to evaluate radioactive contamination and for further analysis at all experimental sites (OLKH, ROZH, K1, K2, K3, and K4). All the studied samples were in the same stages of development at the time of collection: R5—reproductive seed (seeds are tied but do not ripen in the conditions of the middle band), T6—mature tubers, and the peel is light brown [24].

Tubers on the sites were selected without disturbing the soil structure in August-September 2021. After harvesting, the tubers were stored in the soil under natural radiation background conditions in the laboratory at room temperature. The analysis of enzymatic activity and determination of the content of low molecular weight antioxidants were started within a week after the samples were received in the laboratory. To measure the content of ^137^Cs in the soil, samples were taken at each site in a layer of 0–5 cm. Soil samples were stored at room temperature in a tightly closed container until further measurements.

### 2.3. Determination of ^137^Cs in Soils and Plants

The measurements were carried out according to Russian National Standard R 54038-2010, 2012 on the gamma-spectrometer (“Wizard 2480”, Perkin Elmer, Waltham, MA, USA). Soil samples were dried to a dry-air state and then passed through a sieve with a pore diameter of 1 mm. Dry-air samples were prepared according to the same method. To verify the radiation purity of the cuvettes, as well as to gauge the background value of the counting rate (CC_background_), they were placed in the device and measured within 24 h. Clean cells were weighed, filled with samples, and re-weighed to find the mass of the sample (ms_ample_). The filling of the count vials was performed using a sealing piston and in a few steps for maximum densification of the material. This was done to increase the mass of the sample and, accordingly, to increase the accuracy of measurements. The volume of the vials used in the measurements was 24 cm^3^. The mass of the samples ranged from 22.35 to 33.64 g; the experiment was carried out in four repetitions. To improve the accuracy of the results, the measurement time of the sample has been brought almost to the maximum—3600 s (60 min) for each sample. After the specified measurement time, the value of the counting rate (Sample) was obtained. The calculation of the activity of radiocesium (A ^137^Cs) was carried out according to the formula:A ^137^Cs (Bq/kg) = ((CC_sample_ (pulses/h) − CC_background_ (pulses/h))/(1.2 × m_sample_))

### 2.4. Determination of Antioxidant Enzymes Activity

#### 2.4.1. Antioxidant Enzyme Extraction from Tubers

To determine the activity of catalase (CAT), ascorbate peroxidase (APX), and peroxidase (POX) enzymes, *H. tuberosus* tubers (3 tubers from each site) were cleaned from the outer integumentary shells under a stream of cold water, 500 mg of tubers were homogenized separately in 2 mL of 50 mm K-Na-phosphate buffer (pH 7.8) and in 2 mL of 0.2 M Na-acetate buffer (pH 5.0). The obtained homogenates were centrifuged at 12,000× *g* in a centrifuge with cooling (“Heraeus Biofuge Stratos”, Thermo Scientific, USA) for 5 min. The supernatants were stored at 4 °C. Then the extracts were analyzed according to the appropriate methods on a scan-spectrophotometer (“Cary 50 Bio Spectrophotometer UV/V is Reader”, Varian, Palo Alto, CA, USA).

#### 2.4.2. Determination of Catalase Activity

CAT activity was evaluated according to the method described by Goth [24]. A solution containing 1 mL of 65 µM H_2_O_2_ in a 60 mM K-Na-phosphate buffer (substrate) was used for assessment. After 60 s after the addition of the enzyme extract (in a volume of 200 µL), the reaction was stopped by adding 1 mL of 32.4 mM ammonium molybdate ((NH_4_)_6_Mo_7_O_24_·4H_2_O), spectrophotometry was performed at a wavelength of 410 nm, and the adsorption value of the sample (A_sample_) was obtained. Also, to estimate the activity of the enzyme, mixtures A1, A2, and A3 were measured at the same wavelength. The A1 mixture contained 1 mL of the substrate, 1 mL of ammonium molybdate, and 200 µL of enzyme extract. The A2 mixture contained 1 mL of the substrate, 1 mL of ammonium molybdate, and 200 µL of 60 mM K-Na-phosphate buffer. The absorption value of the A3 mixture was achieved by mixing 1.2 mL of 60 mM K-Na-phosphate buffer and 1 mL of ammonium molybdate. The catalase activity was calculated according to the following formula:Catlase Activity (kU/L) = ((A_sample_ − A1) · 271)/(A2 − A3).

Additionally, a quantitative measurement of the protein content in the enzyme preparation was carried out using a qualitative reaction with the dye Coomassie Brilliant Blue G250 [25].

#### 2.4.3. Determination of Ascorbate Peroxidase Activity

APX activity was evaluated according to methodological recommendations by Nakano and Azusa [26]. A solution containing 1.5 mL of 50 mM K-Na-phosphate buffer, 500 µL of 0.5 mM ascorbic acid solution, 500 µL of 0.1 mM hydrogen peroxide solution, and 100 µL of 0.1 mM EDTA solution was used for evaluation.

Measurements were taken every 10 s for 2–3 min after the addition of the enzyme extract (400 µL) at a wavelength of 290 nm.

The control sample contained 1.5 mL of 50 mM K-Na-phosphate buffer, 1 mL of distilled water, 400 µL of enzyme extract, and 300 µL of 0.1 mM EDTA.

Measurements in the control sample are carried out in the same way as measurements in the experimental sample. Measurements in both samples were taken three times. Additionally, a quantitative measurement of the protein content in the enzyme sample was carried out using a qualitative reaction with the dye Coomassie Brilliant Blue G250 [25].

#### 2.4.4. Determination of Peroxidase Activity

POX activity was evaluated according to the methodology described by Gibson et al. with modifications [27]. The following solution was used for evaluation: 1 mL of 0.2 M Na-acetate buffer (pH 4.9), 500 µL 4 µM indigo carmine solution, and 500 µL 0.03 M hydrogen peroxide solution.

Measurements were taken every 15 s for 2 min after the addition of the enzyme extract (500 µL) at a wavelength of 610 nm.

For the control sample, the same volume of distilled water was added to it instead of the enzyme extract.

Additionally, a quantitative measurement of the protein content in the enzyme preparation was carried out using a qualitative reaction with the dye Coomassie Brilliant Blue G250 [25].

### 2.5. Determination of Low Molecular Weight Antioxidants Concentrations

#### 2.5.1. Low Molecular Weight Antioxidant Extraction from Tubers

To determine the content of the sum of water- and alcohol-soluble phenolic compounds, the sum of water- and alcohol-soluble antioxidants, proline, ascorbic acid, reducing sugars, and flavonoids, tubers were cut into thin slices, pre-frozen at −20 °C for 24 h, then placed in a freeze-drying unit (“BenchTop K”, VirTis, Los Angeles, CA, USA) for 24 h. After that time, the plant material was grounded using a mortar and pestle. Lyophilizates were stored at 4 °C in a tightly closed container until the experiments were performed.

#### 2.5.2. Determination of Total Antioxidant Activity

The antioxidant power of the aqueous and lipid fractions of the sum of antioxidants was evaluated according to the method presented in the work by Prieto et al. [28]. Extraction was conducted using distilled water and 80% ethanol (EtOH), respectively, at the ratio of raw materials:extractant—1:5 for 1 h in the dark with constant stirring. After that time, the solutions were centrifuged at 10,000 rpm for 15 min. The supernatant was used for analysis. To determine the content of the aqueous and lipid fractions of antioxidants, a solution containing 200 µL of extract, 2 mL of phosphoric-molybdenum reagent (0.6 M sulfuric acid, 28 mm Na_3_PO_4_, 4 mM (NH_4_)_6_Mo_7_O_24_·4H_2_O) was used. The samples were incubated for 90 min at 95 °C. The measurement was performed at 695 nm. The control sample contained the same amount of extractant instead of the extract. Additionally, curves were constructed based on different concentrations of standard solutions—aqueous L-ascorbic acid and alcohol acetate α-tocopherol.

#### 2.5.3. Determination of Total Water-Soluble and Alcohol-Soluble Phenolics Content

The content of the sum of water- and alcohol-soluble phenolic compounds was evaluated by the Folin-Ciocalteu reagent method [29].

Water-soluble phenolics were extracted using distilled water at the ratio of raw materials:extractant—1:10 at 65 °C for 15 min. The solutions were filtered and used for analysis. To assess the content of water-soluble phenolic compounds, a solution containing 250 µL of extract, 250 µL of Folin-Ciocalteu reagent, 500 µL of 2 M Na_2_CO_3_ solution, 4 mL of extractant (in distilled water) was used. The resulting mixtures were kept in the dark for 25 min and measured at 725 nm against the control—a solution of the same composition as the experimental sample, except distilled water, was added instead of the extract.

The extraction and measurement of the amount of alcohol–soluble phenolic compounds were carried out according to the same method as water-soluble phenolic compounds and differed only in the extractant—80% EtOH was used instead of distilled water.

#### 2.5.4. Determination of Total Flavonoid Content

The content of the flavonoids was evaluated by the Aluminum Chloride Assay [30].

Flavonoids were extracted with 96% EtOH at the ratio of raw materials:extractant—1:10. The samples were placed in a dark container and extracted for 24 h with constant stirring. Then they were centrifuged at 5000 rpm for 5 min. The supernatant was taken and used for analysis. To determine the content of flavonoids, the following solution was used: 1 mL of extract, 500 µL of 1.2% AlCl_3_ EtOH solution, and 500 µL of 120 mM CH_3_COOK. The samples were kept in the dark for 30 min and measured at 425 nm against the control. The control mixture had the same composition as the experimental one, but 96% EtOH was added in the same volume instead of the extract.

#### 2.5.5. Determination of Proline Concentration

The assessment of the proline content was performed [31].

Extraction was carried out with hot distilled water at the ratio of raw materials:extractant—1:100 in a water bath for 10 min at a temperature of 100 °C. Then the extracts were centrifuged at 5000 rpm for 5 min. The supernatants were used for analysis. To measure the proline content, a solution was used: 2 mL of glacial acetic acid (water-free acetic acid), 2 mL of ninhydrin reagent prepared without heating (1.25 g of ninhydrin, 30 mL of glacial acetic acid, 20 mL of 6M H_3_PO_4_), 2 mL of extract. The samples were incubated in a water bath for 20 min at a temperature of 100 °C and then quickly cooled to room temperature. The measurement was made at a wavelength of 520 nm against the control (a mixture of the same composition as for the experimental sample, but distilled water was added in the same volume instead of the extract).

The proline content was found by the calibration curve, using chemically pure proline for its design.

#### 2.5.6. Determination of Ascorbic Acid Content

The ascorbic acid content was measured [32]. Extraction was carried out using 40% and 96% EtOH at the ratio of raw materials:extractant—1:20 in a water bath for 30 min at a temperature of 50 °C. The extracts were filtered and used for analysis. To measure the content of ascorbic acid, a solution containing 2 mL of extract and 2 mL of phosphorus-molybdenum reagent (2.1 mM NaH_2_PO_4_, 3.4 mM (NH_4_)_6_Mo_7_O_24_·4H_2_O) was used.

The samples were incubated in a water bath for 10 min at a temperature of 100 °C, then cooled under a stream of cold water. The measurement was carried out at a wavelength of 730 nm against the control—distilled water.

#### 2.5.7. Determination of Reducing Sugars

The content of reducing sugars was assessed as described previously [33].

Reducing sugars were extracted using distilled water at the ratio of raw materials:extractant—1:50. Extraction was carried out in a water bath for 30 min at 100 °C. The extracts were filtered and used for analysis. To determine the content of reducing sugars, a solution containing 2 mL of solution 1 (69.2 mM CuSO_4_), 2 mL of solution 2 (177.2 mM KNaC_4_H_4_O_6_·4H_2_O, 10.9 mM K_4_[Fe(CN)_6_], 1.86 M NaOH) and 2 mL of extract was used. The samples were incubated in a water bath for 3 min at 100 °C. Then the mixture was immediately transferred to the cell of a spectrophotometer, and adsorption was measured at a wavelength of 670 nm. The content of reducing sugars was calculated using a calibration curve constructed based on chemically pure D-glucose.

### 2.6. Study of Climatic Conditions

The study examined changes in air temperature and precipitation. We analyzed the datasets from variations in climatic zones with varying climatic conditions provided by the All-Russian Scientific Research Institute of Hydrometeorological Information [34]. To characterize these zones, we used data by changes in the monthly average ambient temperature and atmospheric precipitation. Data from 4 different weather stations are presented in Table 2.

### 2.7. Statistical Analysis

Statistical data analysis was performed with the methods of parametric and nonparametric statistics in Microsoft Office Excel (Version 16.33, Washington, DC, USA) and Prism 9 (Version 9.5.1, Boston, MA, USA) software. To calculate the activity of enzymes, the spectrophotometer readings were converted into units of enzyme activity—ncat/g protein. To calculate the content of low-molecular antioxidants, the spectrophotometer readings were converted into mg/gd dry weight. Data on changes in weather conditions are presented in the form of monthly averages in international units of measurement—degrees Celsius (°C) and millimeters of mercury (mm) for ambient air temperature and precipitation, respectively. The data is presented as “average value ± standard error”. The significance of the distinctiveness was assessed using univariate analysis of variance (ANOVA) in combination with Tukey’s test. To assess the relationship between the activity of radiocesium, the activity of enzymes, and the content of low molecular weight antioxidants, the method of principal components analysis (PCA) was used.

## 3. Results

### 3.1. Determination of ^137^Cs in Soils and Plants

The estimation of the ^137^Cs radionuclide content in the soil showed that the experimental sites differed in the level of radioactive contamination: the activity of ^137^Cs in the soils of the contaminated sites was much higher than the activity at the control sites (Table 2).

Considering the effects of radiation on biota, it is important to consider not only external but also internal radiation because gamma radiation produced by ^137^Cs is highly penetrating. In the conducted experiment, a low transition coefficient (TC) ^137^Cs was observed in the soil-plant system. Table 3 shows the content of radiocesium in the soil and in Jerusalem artichoke tubers, as well as the transition coefficients. The classification of soils according to the level of radioactive contamination is provided according to the zoning of territories by the Law of the Russian Federation “On Social Protection of citizens exposed to radiation as a result of the Chornobyl disaster”, 1991.

The control sites (K1, K2, K3, and K4) are characterized by background values of radioactive contamination, while the sites on the territory of the Bryansk region (ROZH and OLKH) are characterized by a low level of contamination.

### 3.2. Determination of Antioxidant Enzymes Activity

The activity of antioxidant enzymes in 6 experimental sites turned out to be relatively heterogeneous. The average indicators of enzyme activity are presented in Table 4. The activity of ascorbate peroxidase (APX) in control samples (K1-K4) of plants varied from 198.51 to 505.12 ncat/g protein, while in samples growing in contaminated areas, it was significantly higher and ranged from 733.87 to 821.2 ncat/g protein for OLKH and POZH respectively. At the same time, the activity of catalase (CAT) and peroxidase (POX) in plants growing in contaminated areas remained at the level of control samples.

#### 3.2.1. Determination of Catalase Activity

The results of catalase activity determination are shown in Figure 2. There was no correlation found between catalase activity and radiation exposure, and there were also no significant differences detected in enzyme activity in control and experimental samples, indicating that there were no changes in the activity of this enzyme in the tubers of *H. tuberosus* plants when grown under conditions of increased radiation background.

#### 3.2.2. Determination of Ascorbate Peroxidase Activity

The results of determining APX activity are shown in Figure 3.

It was shown that APX activity strongly correlates positively with radiation exposure. We also found a significant difference between the control and experimental variants, which may indicate the participation of this enzyme in the tubers of *H. tuberosus* plants in changes occurring under constant exposure to elevated radiation background.

#### 3.2.3. Determination of Peroxidase Activity

The results of determining POX activity are shown in Figure 4.

The activity of POX weakly negatively correlates with the level of radiation exposure. No significant differences were found between the experimental variants compared to the control ones, which indicates that there are no changes in the activity of this enzyme in the tubers of *H. tuberosus* plants in response to growth with increased radiation background.

### 3.3. Determination of Low Molecular Weight Antioxidants Concentrations

The content of low molecular weight antioxidants in 6 experimental sites turned out to be very heterogeneous. The average concentrations of low molecular weight antioxidants are shown in Table 5. The levels of water- and alcohol–soluble phenolic compounds were higher in the areas with radioactive contamination, while in the control areas, they remained relatively the same—from 0.48 to 0.71 μg/mg DW for water-soluble phenolic compounds (WPC) and from 0.258 to 0.510 μg/mg DW for alcohol-soluble phenolic compounds (APC) responsibly. The concentrations of water–soluble antioxidants remained at the same level at all sites—from 0.26 to 0.30 g/mg DW, and the concentrations of lipid-soluble antioxidants remained at the same level (from 0.55 to 0.64 μg/mg DW) except for K4, for which this indicator was 0.27 μg/mg DW. The concentrations of proline (PRO), reducing sugars (RS), and flavonoids (FL) remained at the same level, while the concentration of ascorbic acid (AsA) in the sites with radioactive contamination was significantly higher—by 10–40 times relative to the control sites and amounted to 0.02089 and 0.01864 μg/mg DW for OLKH and ROZH accordingly.

#### 3.3.1. Determination of Total Antioxidant Activity

Indices of the aqueous and lipid fractions of the sum of antioxidants reflect the content of the pool of substances in each of the fractions with antioxidant properties. The results of this analysis are presented in Figure 5.

The data obtained reflect a weak response of the antioxidant system in general. However, despite this, we can see variations in individual components of the antioxidant system. Thus, for all variants, the concentration of water-soluble antioxidants was lower than fat-soluble ones (except for the K4 variant).

#### 3.3.2. Determination of Total Water-Soluble and Alcohol-Soluble Phenolics Content

The antioxidant properties of phenolic compounds are determined by their structure and depend on the number of hydroxyl groups and their position, the presence of a double bond (C2=C3), glycosylation, as well as the presence of substituents in the rings [35]. Plant phenolic compounds have numerous hydroxyl groups and, therefore, can absorb many free radicals.

The measurements of the total content of water-soluble and alcohol-soluble phenolic compounds are shown in Figure 6.

Despite the weak level of correlation between radiation exposure and the amount of alcohol-soluble phenolic compounds, there was a significantly significant increase in their level in the OLKH variant compared to the control variants (*p* < 0.00001), as well as an average level of correlation between radiation exposure and the amount of water-soluble phenolic compounds. Moreover, in both experimental samples, a significant increase in the concentration of water-soluble phenolic compounds was observed compared to the control variants (*p* < 0.00001 and *p* < 0.5 in OLKH and ROZH, respectively). Based on our data, it can be assumed that phenolic compounds take part in changes in the tubers of *H. tuberosus* plants which grow in conditions of increased radiation background.

#### 3.3.3. Determination of Total Flavonoid Content

Flavonoids are compounds of phenolic nature and have antioxidant properties. The antioxidant activity of flavonoids is provided by the inhibition of oxidative enzymes, direct capture of free radicals or ROS, and chelating properties. The results of determining the total content of flavonoids in Jerusalem artichoke tubers are shown in Figure 7.

No correlations were found between the content of phenolic compounds in the samples, and the level of radiation exposure, as well as no significant differences, were found between the control and experimental samples, indicating that there were no changes in the concentrations of flavonoids in the tubers of *H. tuberosus* plants in response to growth with an increased radiation background.

#### 3.3.4. Determination of Proline Concentration

Proline is a non-specific antioxidant that exhibits mainly osmoprotective properties. The results of the proline measurement are shown in Figure 8.

The proline content is weakly negatively correlated with radiation exposure. There were also no significant differences between the concentrations of proline in the experimental and control samples, which indicates that there is no change in the concentrations of proline in the tubers of *H. tuberosus* plants grown under conditions of increased radiation background.

#### 3.3.5. Determination of Ascorbic Acid Content

Ascorbic acid is considered the most common antioxidant capable of neutralizing many types of ROS. The results of determining the concentration of ascorbic acid are shown in Figure 9.

A strong positive correlation was found between the level of ascorbic acid and radiation exposure and significant differences (*p* < 0.00001), which indicates the response of the ascorbate-glutathione cycle, which maintains the ascorbate content in the tubers of *H. tuberosus* plants grew in conditions of increased radiation background.

#### 3.3.6. Determination of Reducing Sugars

Reducing substances are mainly represented by soluble carbohydrates, which, it is considered, can participate in response to radiation both directly by inactivating the hydroxyl radical and indirectly by regulating the expression of genes associated with glutathione metabolism. The results of determining the amount of reducing sugars in Jerusalem artichoke tubers are shown in Figure 10.

Despite the average positive level of correlation between reducing sugars and the level of radiation exposure, no significant differences were found in the concentrations of reducing substances between the experimental and control variants, which indicates that there is no change in the concentrations of reducing substances in the tubers of *H. tuberosus* plants when growing in conditions of increased radiation background.

### 3.4. Changes in Climatic Conditions

Ambient air temperature and precipitation are the main environmental factors that can cause various stressful conditions and change the antioxidant status of plants. Data on the variation of air temperature and precipitation for each month in the period from 2017 to 2021 are presented in the Appendix A (Appendix A).

According to Francesco et al., [36] *H. tuberosus* is considered a good crop for areas with 500 mm of precipitation per year or higher. In our case, the amount of precipitation in the period from 2017 to 2021 ranged from 429 to 807.3 mm. for OLKH and ROZH. Moreover, 2021 was characterized by the largest amount of precipitation—807.3 mm. Also, for K1 and K2, precipitation totaled from 553.3 to 869.8 mm, for K3—from 536.3 to 662.2 mm, and for K4—from 507.8 to 705.1 mm. *H. tuberosus* is considered a good crop for areas with a sum of positive temperatures above 3300 °C. In our case, in the period from 2017 to 2021, the sum of positive temperatures was from 3069.3 to 3329.9 °C for OLKH and ROZH, as well as for K1 and K2, the sum of positive temperatures was from 2686 to 3168.8 °C, for K3-from 2966.7 to 3223 °C, for K4—from 2676.1 to 3089.3 °C.

Thus, according to the presented materials, we observe a sufficient level of moisture and temperature for all the studied areas, which allows us to conclude that the plants under study were not in a state of heat, cold, or drought stress.

### 3.5. Statistical Processing

Analyzing all the measured parameters of the antioxidant system in the aggregate, the resultant ordination plot showed a clear distribution of samples of the growing material in polluted and non-polluted areas (Figure 11). Two main components (PC1 and PC2) explain 56.79% of the total change in the content of antioxidant compounds and show a clear separation of the studied populations of *H. tuberosus* K1, K2, K3, K4 relative to OLKH and ROZH by PC1, as well as the separation of K4 and POZH relative to OLKH by PC2.

The resulting scattering graph showed the distribution of changes in the content of antioxidant compounds and the activity of antioxidant enzymes and revealed a positive correlation between APC, LTA, WPC, ASA, and PRO and a negative correlation of POX with these variables. Similarly, CAT, WTA, and PRO had a positive correlation. The content of ^137^Cs and APX had a positive correlation, and RS did not correlate with any of the parameters. In general, biplot PCA showed that the changes in the parameters of the antioxidant system were greater relative to spatial fluctuations. Thus, abiotic loads in the form of gamma radiation seem to have a significant effect on the antioxidant system of *H. tuberosus* plants.

## 4. Discussion

As a result of anthropogenic impact (accidents at nuclear facilities, nuclear weapons testing), radionuclides such as ^131^I, ^137^Cs, and ^134^Cs and others enter the environment. Among the radionuclides accumulated in the environment, primarily in soils, the most dangerous и prevalent is ^137^Cs, since it has a relatively long half-life compared to other radioactive substances [37]. In the territories affected by the Chornobyl accident, the main dose-forming radionuclide is ^137^Cs, according to the report presented at the Chornobyl Forum [23]. The intensity of contamination with this radionuclide in 2020 in the territories of the Bryansk region, where the experimental sites are located, was 1–5 Cu/km^2^ [38]. In our study, it was shown that the experimental areas exposed to radionuclides after the accident at the Chornobyl NPP could be characterized as low-contaminated by ^137^Cs content in the soil. However, this level significantly exceeds the control background level (Table 2), which indicates a long-term low-dose effect of ionizing radiation on plants growing in the area.

Ionizing radiation exposure to the biological system activates a series of signals that begin with the absorption of energy and eventually lead to damage [39]. The main target upon exposure to ionizing radiation is water molecules present in all organisms. The primary reactions are excitation and ionization, resulting in the formation of ionized water molecules (H_2_O^•+^), H_2_O_2,_ and OH^•^ radicals [40]. In living organisms, this type of ionization starts chain reactions that produce secondary ROS due to the capture of H^•^ and e^−^. The OH^•^ radicals actively interact with the cell’s macromolecules, including lipids, proteins, and DNA, damaging them [41].

Therefore, changes in proteins and carbohydrate metabolism in plants growing under chronic ionization exposure can be related to free radicals’ impact. Often these metabolic changes are related to antioxidant systems directly involved in the neutralization of free radicals [42,43]. Gamma radiation’s effect on antioxidant defense enzyme activity is ambiguous and is probably determined by exposure time, dose, plant species, and other concomitant factors.

For example, it was shown in Arabidopsis plants that exposure to gamma radiation led to a decrease in the activity of catalase and peroxidase. The authors of the study conclude that responses and activation of antioxidant systems in plant organisms depend on the power and dose of radiation [42].

The study of chronically exposed populations of Scots pine in the Chornobyl NPP zone showed that radiation exposure with a dose rate above 50 mGy caused oxidative stress and led to an increase in the concentration of antioxidants in the studied populations. Dose rates over 10 mGy caused an increased frequency of mutations and changes in the genetic structure of Scot’s pine populations. However, the same doses did not affect enzymatic activity [44].

In general, phenolics and flavonoid content in plants increase in response to low doses of gamma radiation exposure and other types of stress. High doses of radiation exposure and prolonged stress are associated with a decrease in phenolics and flavonoid content [45,46,47]. Our study revealed a significant increase in the content of water-soluble phenolics in samples exposed to gamma radiation.

Some available data showed that proline content in seedlings exposed to gamma radiation increased slightly as the dose of gamma radiation was increased [48,49]. However, Falahati et al. [50] disproved this claim by suggesting that radiation could have increased antioxidant levels and, therefore, there would be no need for additional proline to cope with the same level of oxidative stress. Our data are consistent with this statement. Proline content is slightly negatively correlated with the level of exposure to ionizing radiation.

Several data indicate that a single exposure to gamma radiation can reduce the ascorbic acid content in some plant species [51]. Changes in ascorbic acid accumulation are associated with radiation dose and exposure time. Long-term exposure or short-term exposure at low doses can stimulate the accumulation of ascorbic acid in plants [40], which is consistent with our data.

Some studies indicate a decrease in the concentration of soluble sugars in response to ionizing radiation exposure, which is associated with the conversion of sugars to starch and adaptive responses [51,52]. In our study, there were no significant differences in soluble sugar content relative to the controls.

The response of plants to prolonged exposure to ionizing radiation is a complex interaction between radiation dose, dose rate, temporal and spatial variations, different radiation sensitivities of different plant species, and indirect effects from other events. Repeated ionizing radiations, acute or chronic, guarantee the adaptation of plant species exhibiting radioresistance [16].

Recently, only one study has been available on the effect of ionizing radiation on Jerusalem artichoke plants. Low doses effect of ionizing radiation on the growth, yield, and some biochemical parameters of *H. tuberosus* plants was studied. Tubers were subjected to a single exposure of different low doses of gamma irradiation through a ^137^Cs source. A dose of 5 Gy had a pronounced stimulating effect on plant growth, dry matter content, photosynthetic pigments, nitrogen, phosphorus, potassium, total carbohydrates, and inulin. Phenolic and flavonoid content correlated with the dose of ionizing radiation and reached a maximum at doses of 5–10 Gy [53].

## 5. Conclusions

The present data demonstrate that antioxidant responses to chronic low doses of exposure to ionizing radiation on Jerusalem artichoke plants are associated with the increase in the activity of ascorbate peroxidase and catalase, and the increase in the concentration of water-soluble non-enzymatic antioxidants, such as ascorbic acid and phenolic compounds. Different plant species react differently to long-term exposure to ionizing radiation, depending on the dose, exposure time, and other factors. Our results require clarification and additions over the course of several years in order to be able to use this response pattern as a marker of oxidative stress. It would also be interesting to investigate similar reactions in other plant species regularly growing in the studied area. Within the framework of this study, it is also possible to use other methods for assessing the level of oxidative stress, such as determining lipid peroxidation, levels of DNA damage, determining hydrogen peroxide, superoxide anion, etc. These data will make it possible to assess the impact of even low doses of ionizing radiation more accurately on the stability of various ecosystems. Even small doses of ionizing radiation are an environmental factor that must be taken into account.

## Figures and Tables

**Figure 1 antioxidants-12-00956-f001:**
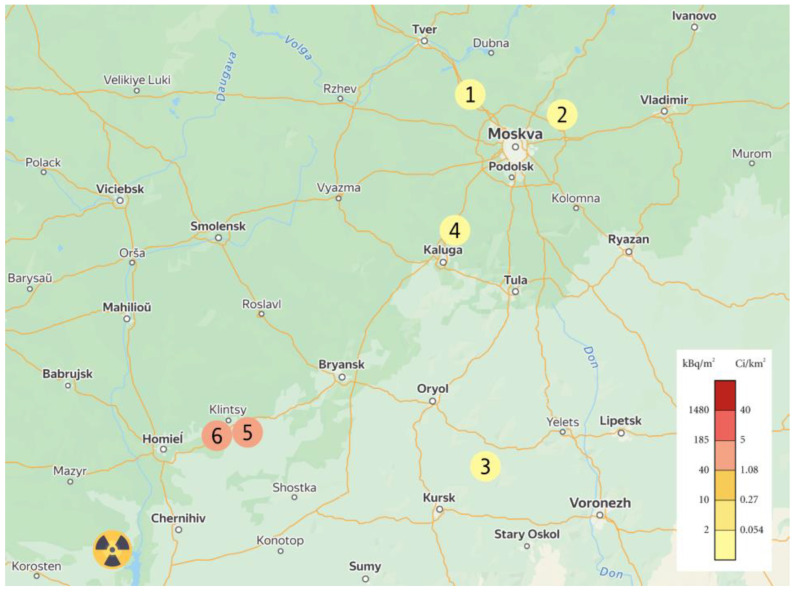
Location of control and experimental areas. The map was created based on the Google Maps service and modified in CorelDraw Graphics Suite 2021. The explanation for numbers in the figure is presented in Table 1. The levels of radioactive contamination are given for 2006 by data from International Atomic Energy Agency (IAEA) [23].

**Figure 2 antioxidants-12-00956-f002:**
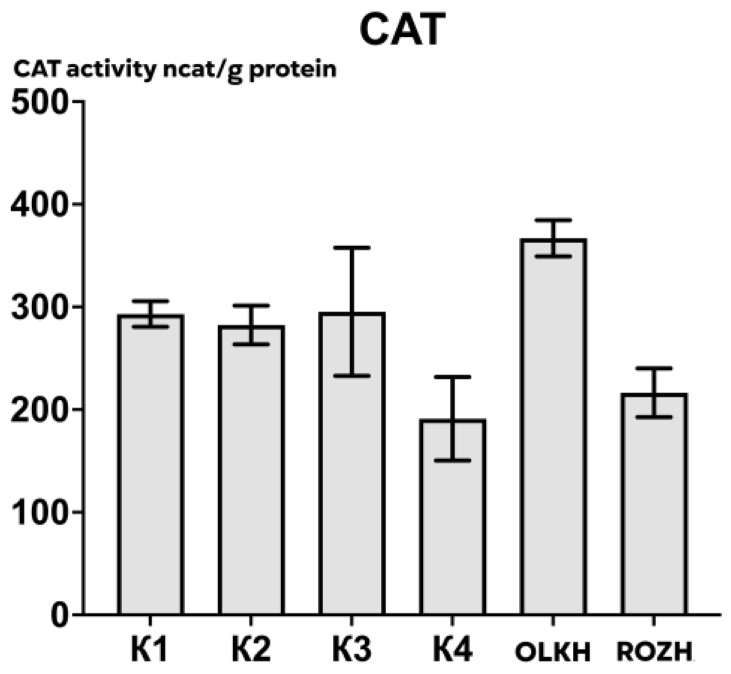
CAT activity in tubers of *H. tuberosus* growing in contaminated areas and in areas without contamination.

**Figure 3 antioxidants-12-00956-f003:**
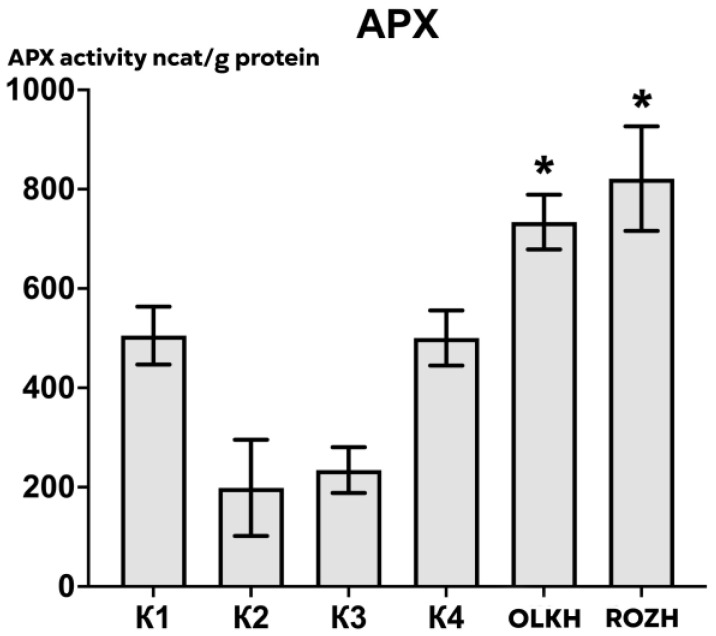
APX activity in tubers of *H. tuberosus* growing in contaminated arias and arias without contamination. * corresponds to *p* < 0.05 significance level.

**Figure 4 antioxidants-12-00956-f004:**
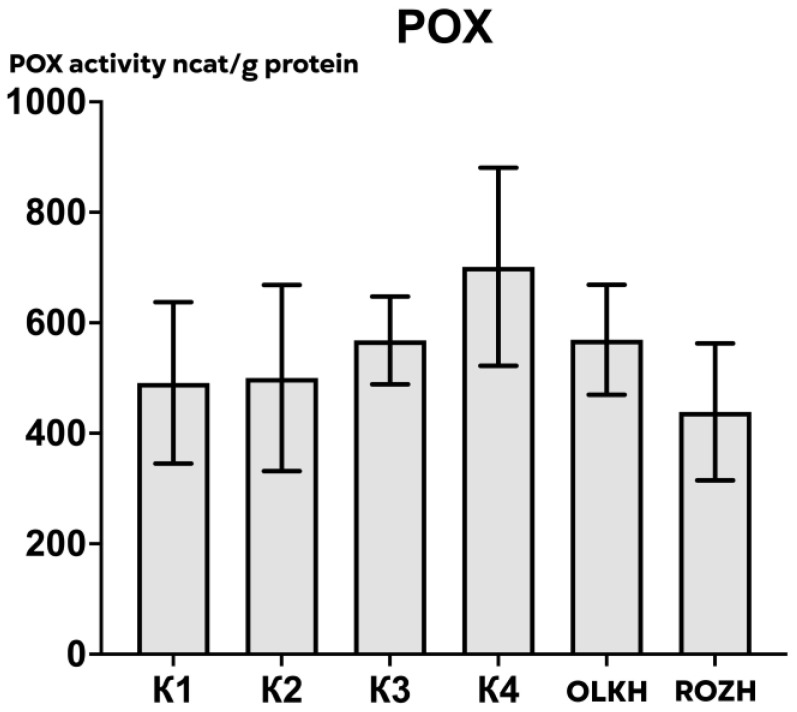
POX activity in tubers of *H. tuberosus* growing in contaminated arias and in arias without contamination.

**Figure 5 antioxidants-12-00956-f005:**
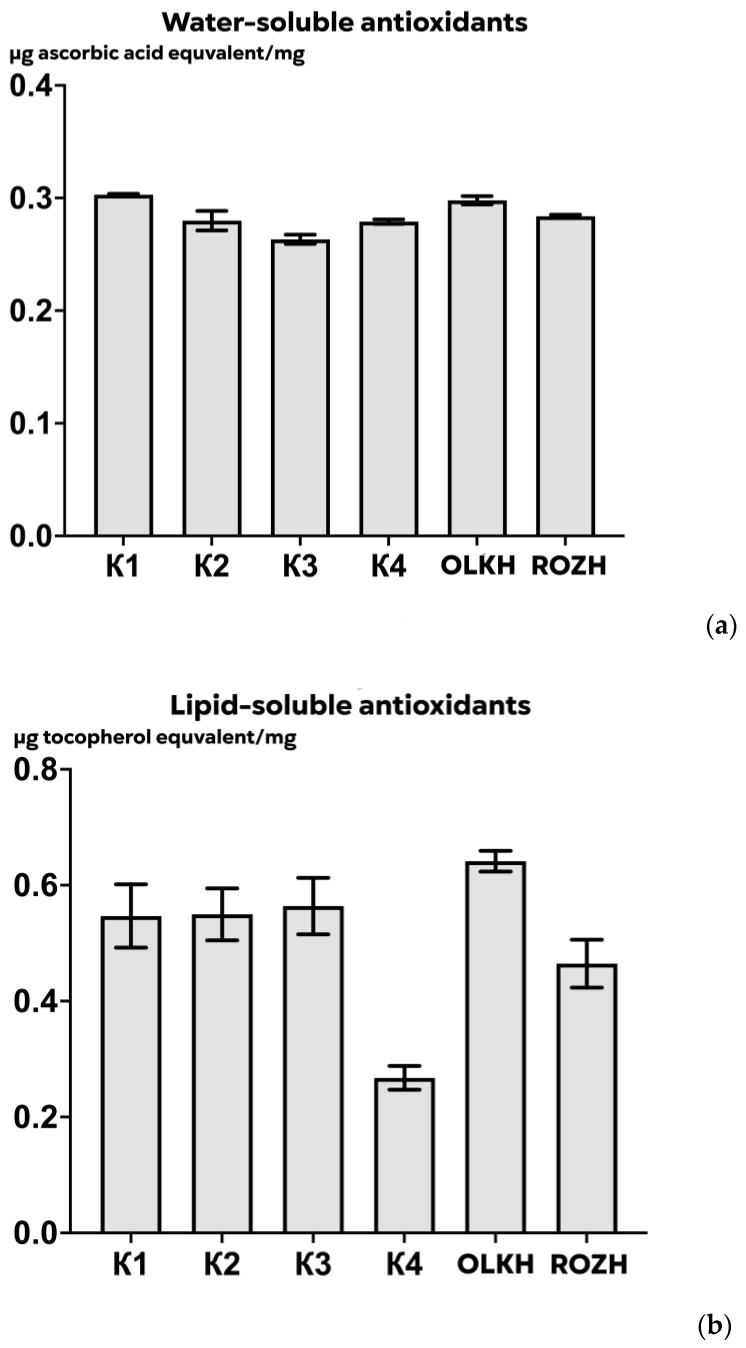
The concentration of water-soluble (**a**) and lipid-soluble (**b**) antioxidants in tubers of *H. tuberosus* growing in contaminated arias and in arias without contamination.

**Figure 6 antioxidants-12-00956-f006:**
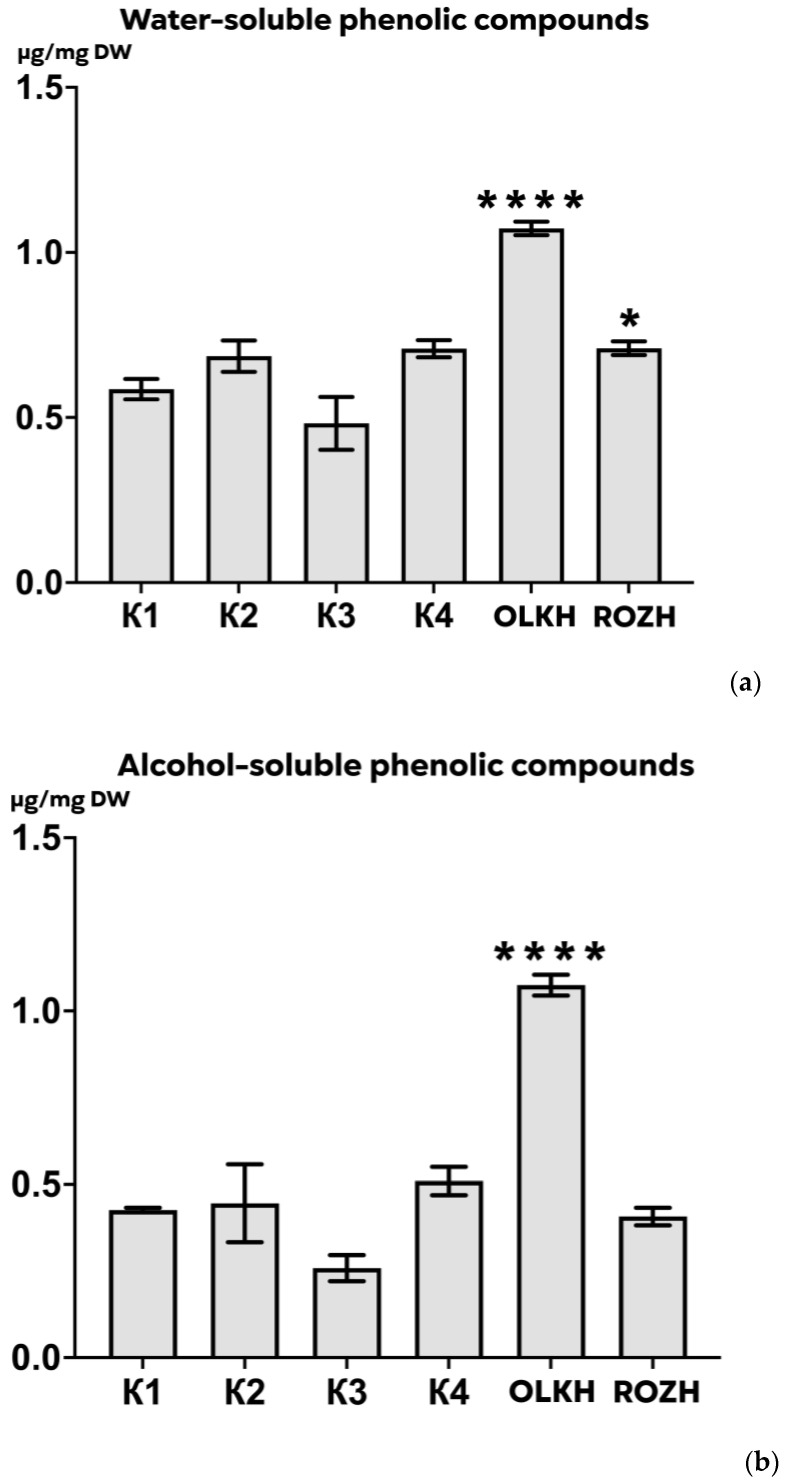
The concentration of water-soluble (**a**) and alcohol-soluble phenolic (**b**) compounds in tubers of *H. tuberosus* growing in contaminated arias and in arias without contamination. **** corresponds to *p* < 0.00001 significance level; * corresponds to *p* < 0.05 significance level.

**Figure 7 antioxidants-12-00956-f007:**
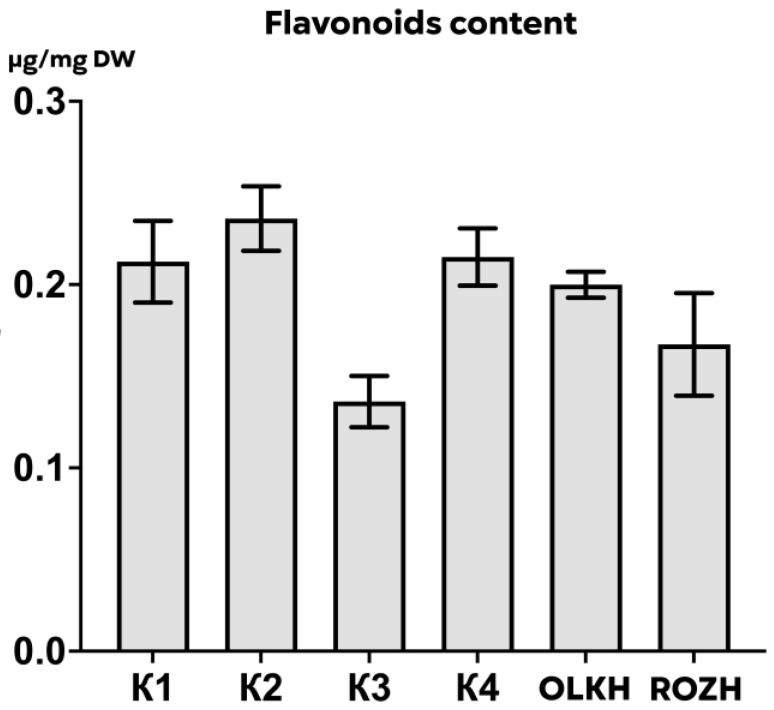
Total flavonoid content in tubers of *H. tuberosus* growing in contaminated areas and in areas without contamination.

**Figure 8 antioxidants-12-00956-f008:**
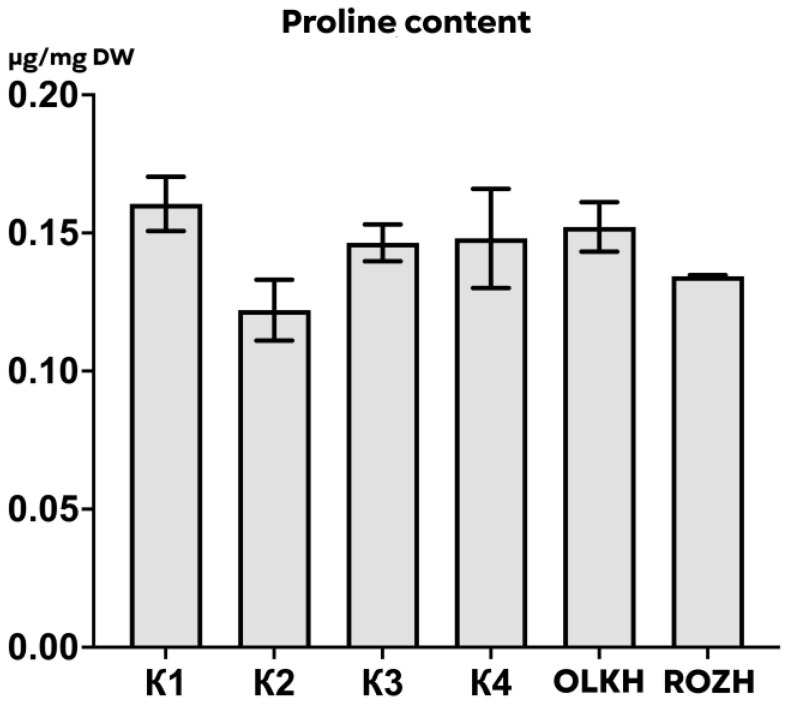
Proline concentration in tubers of *H. tuberosus* growing in contaminated areas and in areas without contamination.

**Figure 9 antioxidants-12-00956-f009:**
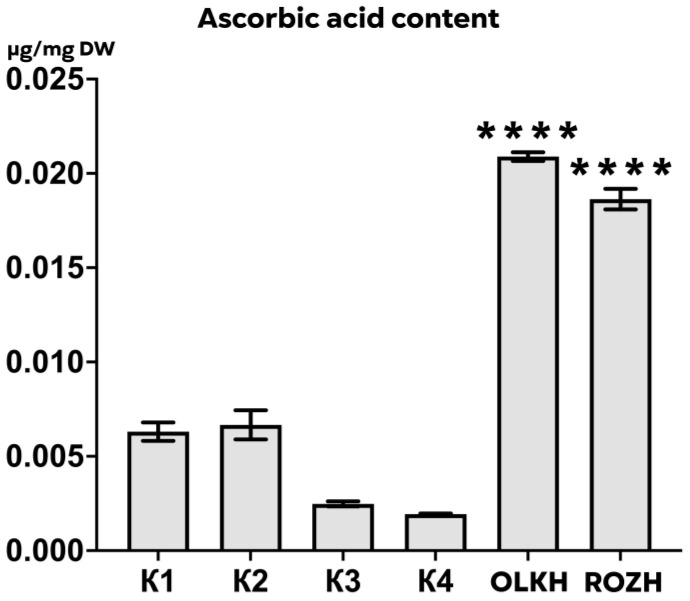
The ascorbic acid content in tubers of *H. tuberosus* grows in contaminated areas and in areas without contamination. **** corresponds to *p* < 0.00001 significance level.

**Figure 10 antioxidants-12-00956-f010:**
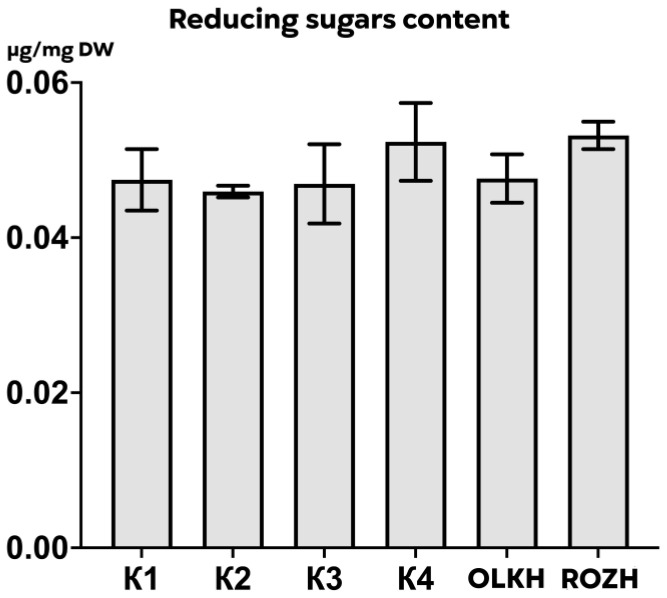
Reducing sugar content in tubers of *H. tuberosus* growing in contaminated areas and in areas without contamination.

**Figure 11 antioxidants-12-00956-f011:**
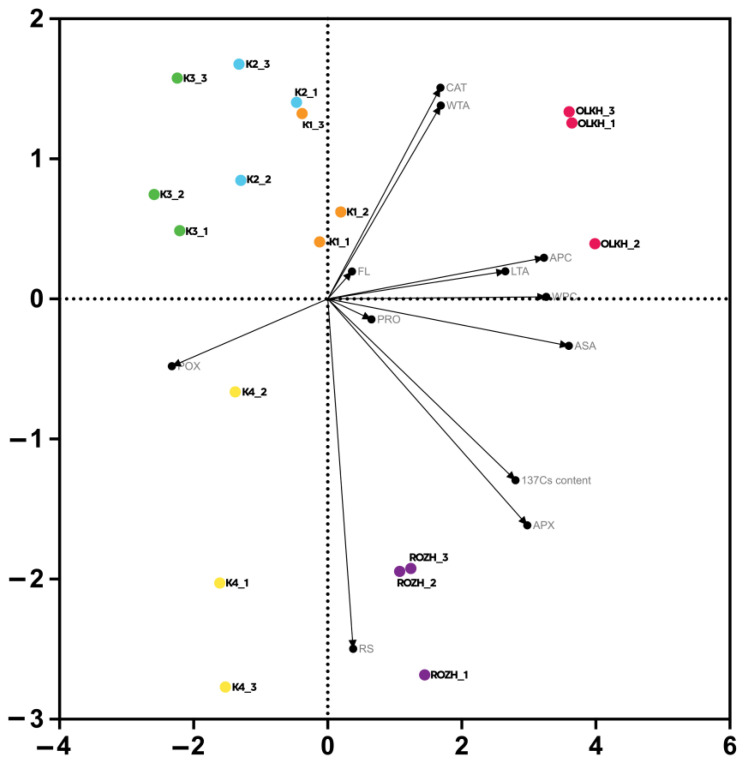
Biplot of the principal component analysis (PCA) for the parameters of the antioxidant system at the sampling sites. Black arrows represent the parameters of the antioxidant system, and circles in color represent sampling points.

**Table 1 antioxidants-12-00956-t001:** Characteristics of experimental areas.

№	Areas	Coordinates	Description
1	K1	Discreet city Solnechnogorsk, 56.121170° N 36.889667° W	Field, 15 m from the road sod-podzolic soil
2	K2	Discreet city Chernogolovka, 56.050380° N 38.423595° W	Field, 50 m from the road sod-podzolic soil
3	K3	Orel region, 52.220666° N 37.020390° W	Field, 20 m from the road chernozemic soil
4	K4	Discreet city Zhukov, 51.709466° N 36.123476° W	Field, 30 m from the road sod-podzolic soil
5	OLKH	Discreet city Klintsy, 52.742743° N 32.130348° W	Field, 50 m from the road sod-podzolic soil
6	ROZH	Discreet city Klintsy, 52.740622° N 32.083456° W	Field, 40 m from the road sod-podzolic soil

**Table 2 antioxidants-12-00956-t002:** Characteristics of weather stations.

№	Areas	Weather Station	Coordinates
1	K1	“Moscow (VDNH)”	Moscow, 55.830286° N 37.623578° W
2	K2		
3	K3	“Ponyri”	Orel region, 52.320755° N 36.311341° W
4	K4	“Kolomna”	Kolomna, 55.137384° N 38.730738° W
5	OLKH	“Krasnya gora”	Discreet city Klintsy, 52.999033° N 31.602508° W
6	ROZH		

**Table 3 antioxidants-12-00956-t003:** ^137^Cs content in soil and *H. tuberosus* plants.

Areas	A ^137^Cs, Bq·kg^−1^ Soil	Density of Surface Contamination, Cu/km^2^	Level of Contamination	^137^Cs Activity in Tubers	TC
K1	23.92 ± 0.91	0.19 ± 0.01	-	Not detriment	-
K2	18.14 ± 1.62	0.15 ± 0.01	-	Not detriment	-
K3	52.23 ± 1.73	0.42 ± 0.01	-	Not detriment	-
K4	21.41 ± 0.72	0.17 ± 0.01	-	Not detriment	-
OLKH	570.78 ± 13.66	4.64 ± 0.03	low	31.96 ± 2.43	0.056
ROZH	350.95 ± 3.40	2.85 ± 0.03	low	18.51 ± 1.13	0.053

**Table 4 antioxidants-12-00956-t004:** The activity of antioxidant enzymes in *H. tuberosus* plants growing in pollution-free zones and in areas with ^137^Cs contamination. * corresponds to *p* < 0.05 significance level.

Areas	APX Activity, ncat/g Protein	CAT Activity, ncat/g Protein	POX Activity, ncat/g Protein
K1	505.12 ± 65.90	293.11 ± 14.18	491.20 ± 165.38
K2	198.51 ± 109.67	282.45 ± 21.32	500.03 ± 190.54
K3	234.34 ± 52.14	295.22 ± 70.53	568.09 ± 89.86
K4	500.49 ± 62.79	191.12 ± 46.04	701.44 ± 202.94
OLKH	733.87 ± 62.34 *	366.96 ± 19.94	569.34 ± 112.58
ROZH	821.20 ± 119.17 *	216.37 ± 26.91	438.61 ± 140.20

**Table 5 antioxidants-12-00956-t005:** The concentration of low molecular weight antioxidants in *H. tuberosus* plants growing in pollution-free zones and in areas with ^137^Cs contamination. **** corresponds to *p* < 0.00001 significance level; * corresponds to *p* < 0.05 significance level.

Areas	WPC, μg/mg DW	APC, μg/mg DW	WTA, μg/mg DW	LTA, μg/mg DW	PRO, μg/mg DW	AsA, μg/mg DW	RS, μg/mg DW	FL, μg/mg DW
K1	0.59 ± 0.03	0.426 ± 0.008	0.30 ± 0.001	0.55 ± 0.06	0.1605 ± 0.0111	0.00631 ± 0.00055	0.047 ± 0.004	0.21 ± 0.03
K2	0.69 ± 0.05	0.445 ± 0.127	0.28 ± 0.010	0.55 ± 0.05	0.1220 ± 0.0125	0.00667 ± 0.00087	0.046 ± 0.001	0.24 ± 0.02
K3	0.48 ± 0.09	0.258 ± 0.043	0.26 ± 0.005	0.56 ± 0.06	0.1464 ± 0.0075	0.00248 ± 0.00015	0.047 ± 0.006	0.14 ± 0.02
K4	0.71 ± 0.03	0.510 ± 0.046	0.28 ± 0.002	0.27 ± 0.02	0.1480 ± 0.0203	0.00193 ± 0.00004	0.052 ± 0.006	0.22 ± 0.02
OLKH	1.07 ± 0.02 ****	1.074 ± 0.034 ****	0.30 ± 0.004	0.64 ± 0.02	0.1522 ± 0.0102	0.02089 ± 0.00026 ****	0.048 ± 0.004	0.20 ± 0.01
ROZH	0.71 ± 0.02 *	0.407 ± 0.029	0.28 ± 0.002	0.46 ± 0.05	0.1343 ± 0.0005	0.01864 ± 0.00061 ****	0.053 ± 0.002	0.17 ± 0.03

## Data Availability

The data is contained within the manuscript and Appendix A.

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
