# Peer review of "Influence of Increased Radiation Background on Antioxidative Responses of Helianthus tuberosus L."

_antioxidants, 2023, doi:10.3390/antiox12040956_

Round 1
Reviewer 1 Report (Previous Reviewer 2)
I have read carefully the revised manuscript and I consider that the Authors have improved the manuscript based on all the comments indicated by the referees. According to that, I recommend its publication in its current form.
Author Response
We are grateful to the reviewer for the recommendation our manuscript to publication.Reviewer 2 Report (Previous Reviewer 1)
Authors sufficiently improved the content of the MS. I would like to recommend it for publication, Whoever authors can further improve the study by incorporating limitations of the study.
Author Response
We are grateful to the reviewer for the recommendation our manuscript to publication. In conclusion we pointed out the limitations of our study
Reviewer 3 Report (New Reviewer)
The manuscript is interesting and novel, however, it is necessary to attend to the following:
· Page 1: the line number was not included throughout the manuscript; it is necessary to include it to facilitate the review
· Abstract line 6: scientific names must be appeared in italic text format
· Introduction paragraph two: was the reference omitted?
· Introduction paragraph seven, line 1: modify… H. tuberosus has great…delete L. (from here on)
· Introduction paragraph seven: was the reference missed?
· Use capital text format in the first letter of each word, only in subsections, i.e.
2.1. Characteristics of Experimental Areas and Plant Material
2.2. Soil and Plant Sample Collection on Experimental Area
2.3. Determination of 137Cs in Soils and Plants
2.4. Determination of Antioxidant Enzymes Activity
2.5. Determination of Low Molecular Weight Antioxidants Concentrations
2.6. Study of Climatic Conditions
2.7. Statistical Analysis
· Materials and Methods section 2.3., line 6: use 24 h instead 24 hours
· Materials and Methods section 2.4.1, line 4: use mL instead ml; correct through the manuscript
· Materials and Methods section 2.4.1, line 6: use min instead minutes; correct were necessary
· Materials and Methods section 2.4.1, line 6: delete space 4 °C
· Materials and Methods section 2.4.1, line 6: indicate the information of the equipment used, ie., centrifuge (model, brand, country)
· Materials and Methods section 2.4.1, line 7,8: indicate the information of the equipment used, ie., spectrophotomer (model, brand, country)
· Materials and Methods section 2.4.2, line 3: use 60 s instead 60 seconds
· Materials and Methods section 2.4.3:
-use 10 s instead 10 seconds
-use 2-3 min instead 2-3 minutes
· Materials and Methods section 2.4.4:
-use 15 s instead 15 seconds
-use 2 min instead 2 minutes
· Materials and Methods section 2.5.1:
-use 24 h instead 24 hours
- indicate the information of the equipment used, ie., freezer-drying (model, brand, country)
· Materials and Methods section 2.5.2:
-use 1 h instead 1 hour
-use 10,000 instead 10000
-use 15 min instead 15 minutes
-use Na3PO4 instead Na3PO4
-use 90 min instead 90 minutes
· Materials and Methods section 2.5.2:
-modify…was evaluated by the Folin-Ciocalteu reagent method [26]
-use 15 min instead 15 minutes
-use 25 min instead 25 minutes
· Materials and Methods section 2.5.4:
-modify…flavonoids was evaluated by the INDICATE THE METHOD NAME [27]
-96% EtOH at…
-use 24 h instead 24 hours
-use 5 min instead 5 minutes
-use 30 min instead 30 minutes
· Materials and Methods section 2.5.5:
- The assessment of the proline content was performed [28]. Extraction was…
-use 10 min instead 10 minutes
-use 20 min instead 20 minutes
· Materials and Methods section 2.5.6:
- The ascorbic acid content was measured [29]. Extraction was…
-use 30 min instead 30 minutes
-use 10 min instead 10 minutes
· Materials and Methods section 2.5.6:
- The content of reducing sugars was assessed as described previously [30]. Reducing sugars were…
-use 30 min instead 30 minutes
· Materials and Methods section 2.6:
- delete space…and precipitation. We analyzed…
· Results:
- use bold text format for 3. Results
· Use capital text format in the first letter of each word, only in subsections, i.e.
3.1. Characteristics of Plant Material
3.2. Determination of 137Cs in Soils and Plants
3.3. Determination of Antioxidant Enzymes Activity
3.4. Determination of Low Molecular Weight Antioxidants Concentrations
3.6. Changes in Climatic Conditions
3.5. Statistical Processing
Note: review the sequence in the numbering of these subsections
· Results, section 3.1.:
- In the results section, only the relevant information of the results obtained is written, references should not be inserted in this section. However, if it is pertinent to combine the Results and Discussion section, you can do so.
· Results, section 3.2.:
- In the results section, only the relevant information of the results obtained is written, references should not be inserted in this section. However, if it is pertinent to combine the Results and Discussion section, you can do so.
-use (Table 2) instead (table 2)
- use literals to indicate differences between zones (information of Table 2)
Note: it is necessary to combine the Results and Discussion section
· Results, section 3.3.:
- Paragraphs must be written in justified text format
- Delete… with their standard deviations
· Results, Table 4:
- Use a dot instead a comma, ie., use 505.12 ± 68.90 instead 505,12 ± 68,90 (correct though the table)
- use literals to indicate differences between zones
· Results, section 3.3.1 to 3.3.3:
- The information described in the figures is already included in Table 4, why present it again?
· Results, section 3.4.:
- Delete… with their standard deviations
· Results, 3.4. to 3.4.6.:
- If figures 5 to 10 correspond to the information contained in the tables, it is necessary to eliminate the figures since it is not necessary to repeat the information. It is important to work with the information in table 5: include the literals to indicate differences, review the text size format indicated in the authors' guide, or in any case, you could separate the information in table 5 into two tables, 5 and 6 respectively.
· Reference sections:
- Reference 5: Capsicum annuum
- Reference 8: Oryza sativa…Phaseolus mungo
- Reference 14,15,16,17,18,19,26,32,35,51: Helianthus tuberosus…
- Reference 23: authors names missed ?
- Reference 31: incomplete reference?
- Reference 32: correct authors text format
- Reference 43: Trigonella foenum-graecum L.
- Reference 48: Oryza sativa
· Another comments:
- Is it possible to perform any chromatographic analysis on phytochemicals of the samples evaluated?
Round 2
Reviewer 3 Report (New Reviewer)
The comments requested in the previous review were addressed
This manuscript is a resubmission of an earlier submission. The following is a list of the peer review reports and author responses from that submission.
Round 1
Reviewer 1 Report
There are many other environmental factor may also be involved; Like water, soil, air. it will be un justify to refer this antioxidant alteration purely to 137CS isotope. further only one marker is not king enough; needs to select many samples from different plants as well as from human being to assess the real study.
Reviewer 2 Report
General comments
The manuscript reports a study on the enzymatic and non-enzymatic antioxidant protection of Helianthus tuberosus against increased radiation background. Regarding the manuscript, the introduction is fine and including the available literature associated with the topic of the study. In genneral, the experiment was well designed and the methods used for this assessment were described with enough detail. In my opinion, the manuscript presents interesting findings that fall within the scope of Antioxidants. Therefore, I suggest the manuscript for publication after minor revision since there are some few points that require improvement or clarification. The study would have benefited from having more than just one season. The English of the manuscript should be revised and improved by a native speaker (see specific comments).
Specific Comments
Abstract:
Line 13. Change 137CS to 137Cs
Rewrite the following sentences: ‘….the activity of antioxidant defense enzymes, such as catalase and peroxidase, weakly correlated with radiation exposure’ and ‘The activity of ascorbate peroxidase, on the contrary, strongly positively correlated with radiation exposure’.
Introduction:
Line 33. Replace ‘30B years‘ by ‘30 years‘.
Line 38. I suggest to change ‘radio emission’ to ‘radiation’ or ‘radiation emission’.
M&M:
I recommend to indicate the sunchoke variety used in the study.
Lines 209 and 213. Amend ‘Folin-Chkolteu reagent’ and ‘Folin-Chocalteu reagent’ to ‘Folin-Ciocalteu reagent’.
Lines 280-282. Use points instead of commas.
Monthly data of temperature and precipitation for all study sites could be included (in this section or as supplementary material). That information is important to conclude that differences were induced by exposure to ionizing radiation and not to other causes like temperature and rainfall.
Results:
In figure (2, 3, 4…) captions ‘arias’ must be replaced by ‘areas’.